# Buy 4 REINFORCE Samples, Get a Baseline for Free!

**Wouter Kool**
University of Amsterdam
ORTEC
`w.w.m.kool@uva.nl`

**Herke van Hoof**
University of Amsterdam
`h.c.vanhoof@uva.nl`

**Max Welling**
University of Amsterdam
CIFAR
`m.welling@uva.nl`

## Abstract

REINFORCE can be used to train models in structured prediction settings to directly optimize the test-time objective. However, the common case of sampling one prediction per datapoint (input) is data-inefficient. We show that by drawing *multiple* samples (predictions) per datapoint, we can learn with significantly less data, as we freely obtain a REINFORCE baseline to reduce variance. Additionally we derive a REINFORCE estimator with baseline, based on sampling *without* replacement. Combined with a recent technique to sample sequences without replacement using Stochastic Beam Search, this improves the training procedure for a sequence model that predicts the solution to the Travelling Salesman Problem.

## 1 Introduction

REINFORCE (Williams, 1992) is a well known policy optimization algorithm that learns directly from experience. Variants of it have been used to train models for a wide range of structured prediction tasks, such as Neural Machine Translation (Ranzato et al., 2016; Bahdanau et al., 2017), Image Captioning (Vinyals et al., 2015b) and predicting solutions (tours) for the Travelling Salesman Problem (TSP) (Bello et al., 2016; Kool et al., 2019a). As opposed to maximum likelihood (supervised) learning, the appeal of using REINFORCE for structured prediction is that it directly optimizes the test-time performance.

When using REINFORCE, often for each datapoint (e.g. a sentence, image or TSP instance) only a single sample/prediction (e.g. a translation, caption or tour) is used to construct a gradient estimate. From a classic Reinforcement Learning (RL) point of view, this makes sense, as we may not be able to evaluate multiple sampled actions for a state (datapoint). However, from a data point of view, this is inefficient if we *can* actually evaluate multiple samples, such as in a structured prediction setting. Reinforcement Learning with multiple samples/predictions for a single datapoint has been used before (e.g. Shen et al. (2016); He et al. (2016)), but we use the samples as counterfactual information by constructing a (local, for a single datapoint) REINFORCE baseline. A similar idea was applied for variational inference by Mnih & Rezende (2016).

Many structured prediction tasks can be formulated in terms of sequence modelling, which is the focus of this paper. In most sequence modelling tasks, the objective is a deterministic function of the predicted sequence. As a result, duplicate sampled sequences are uninformative and therefore do not improve the quality of the gradient estimate. To solve this problem, we propose to use sampling *without* replacement to construct a better gradient estimate. This is inspired by recent work by Kool et al. (2019b), who introduce Stochastic Beam Search as a method to sample sequences without replacement, and use this to construct a (normalized) importance-weighted estimator for (sentence level) BLEU score. We extend this idea to estimate policy gradients using REINFORCE, and we show how to use the same set of samples (without replacement) to construct a baseline. This way we can leverage sampling without replacement to improve training of sequence models.

In our experiment, we consider the TSP and show that using REINFORCE with multiple samples is beneficial compared to single sample REINFORCE, both computationally and in terms of data-efficiency. Additionally, for a sample size of $4 - 8$ samples per datapoint, sampling without replacement results in slightly faster learning.

## 2 BACKGROUND

### 2.1 REINFORCE

The REINFORCE estimator (Williams, 1992) allows to estimate gradients of the expectation $\mathbb{E}_{y \sim p_{\boldsymbol{\theta}}(y)}[f(y)]$ by the relation:

$$\nabla_{\boldsymbol{\theta}} \mathbb{E}_{y \sim p_{\boldsymbol{\theta}}(y)}[f(y)] = \mathbb{E}_{y \sim p_{\boldsymbol{\theta}}}[\nabla_{\boldsymbol{\theta}} \log p_{\boldsymbol{\theta}}(y) f(y)] \tag{1}$$

If we also have a context, or datapoint, $x$ (such as a source sentence), we may write $p_{\boldsymbol{\theta}}(y|x)$ and $f(y, x)$, but in this paper, we leave dependence on $x$ implicit. Extension of the derived estimators to a minibatch of datapoints $x$ is straightforward.

Typically, we estimate the expectation using samples $y_1, ..., y_k$ and we may reduce variance of the estimator by using a *baseline* $B_i$ that is *independent* of the sample $y_i$ (but may depend on the other samples $y_j, j \neq i$):

$$\nabla_{\boldsymbol{\theta}} \mathbb{E}_{y \sim p_{\boldsymbol{\theta}}(y)}[f(y)] \approx \sum_{i=1}^{k} \nabla_{\boldsymbol{\theta}} \log p_{\boldsymbol{\theta}}(y_i)(f(y_i) - B_i) \tag{2}$$

In practice, often a single sample $y$ is used (per datapoint $x$, as we already have a batch of datapoints) to compute the estimate, e.g. $k = 1$, but in this paper we consider $k > 1$.

### 2.2 THE GUMBEL-TOP-$k$ TRICK AND STOCHASTIC BEAM SEARCH

In this paper, we consider a parametric distribution over discrete structures (sequences). Enumerating all $n$ possible sequences as $y^1, ..., y^n$, we indicate with $y^i$ the $i$-th possible outcome, which has log-probability $\phi_i = \log p_{\boldsymbol{\theta}}(y^i)$ defined by the model. We can use the Gumbel-Max trick (Gumbel, 1954; Maddison et al., 2014) to sample $y$ according to this distribution as follows: let $G_i \sim$ Gumbel (a standard Gumbel distribution) for $i = 1, ..., n$ i.i.d., and let $y = y^{i^*}$, where $i^* = \arg\max_i \{\phi_i + G_i\}$. Then $P(y = y^i) = p_{\boldsymbol{\theta}}(y^i)$. For a proof we refer to Maddison et al. (2014). In a slight abuse of notation, we write $G_{\phi_i} = \phi_i + G_i$, and we call $G_{\phi_i}$ the (Gumbel-) *perturbed log-probability* of $y^i$.

The Gumbel-Max trick can be extended to the *Gumbel-Top-$k$* trick (Kool et al., 2019b) to draw an *ordered sample without replacement*, by taking the top $k$ largest perturbed log-probabilities (instead of just one, the argmax). The result is equivalent to sequential sampling without replacement, where after an element $y$ is sampled, it is removed from the domain and the remaining probabilities are renormalized. The Gumbel-Top-$k$ trick is equivalent to Weighted Reservoir Sampling (Efraimidis & Spirakis, 2006), as was noted by Vieira (2014). The ordered sample is also known as a partial ranking according to the Plackett-Luce model (Plackett, 1975; Luce, 1959).

For a sequence model with exponentially large domain, naive application of the Gumbel-Top-$k$ trick is infeasible, but an equivalent result can be obtained using Stochastic Beam Search (Kool et al., 2019b). This modification of beam search expands the $k$ partial sequences with maximum (Gumbel) *perturbed* log-probability, effectively replacing the standard top $k$ operation by sampling without replacement. The resulting top $k$ completed sequences are a sample without replacement from the sequence model, by the equivalence to the Gumbel-Top-$k$ trick. For details we refer to Kool et al. (2019b).

### 2.3 STATISTICAL ESTIMATION WITHOUT REPLACEMENT

For many applications we need to estimate the expectation of a function $f(y)$, where $y$ is the realization of a variable with a *discrete* probability distribution $p_{\boldsymbol{\theta}}(y)$. When using Monte Carlo (MC) sampling (with replacement), we write $y_i$ to indicate the $i$-th sample in a set of samples. In contrast, when sampling *without replacement* we find it convenient to write $y^i$ (with superscript $i$) to refer to the $i$-th possible value in the domain, so (like we did in Section 2.2) we can enumerate the domain with $n$ possible values as $y^1, ..., y^n$. This notation allows us to write out the expectation of $f(y)$:

$$\mathbb{E}_{y \sim p_{\boldsymbol{\theta}}(y)}[f(y)] = \sum_{i=1}^{n} p_{\boldsymbol{\theta}}(y^i) f(y^i). \tag{3}$$

Using MC sampling with replacement, we estimate equation 3 using $k$ samples $y_1, ..., y_k$:

$$\mathbb{E}_{y \sim p_{\boldsymbol{\theta}}(y)}[f(y)] \approx \sum_{i=1}^{k} f(y_i). \tag{4}$$

When sampling *without replacement* using the Gumbel-Top-$k$ trick (Section 2.2) we write $S$ as the set of $k$ largest indices of $G_{\phi_i}$ (i.e. $S = \arg \operatorname{top} k\{G_{\phi_i} : i \in \{1, ..., n\}\}$), so the sample (of size $k$) without replacement is $\{y^i : i \in S\}$. We can use the sample $S$ with the estimator derived by Vieira (2017), based on priority sampling (Duffield et al., 2007). This means that, to correct for the effects of sampling without replacement, we include importance weights $\frac{p_{\boldsymbol{\theta}}(y^i)}{q_{\boldsymbol{\theta}, \kappa}(y^i)}$. Here $\kappa$ is the $(k+1)$-th largest value of $\{G_{\phi_i} : i \in \{1, ..., n\}\}$, i.e. the $(k+1)$-th largest Gumbel perturbed log-probability, and $q_{\boldsymbol{\theta}, a}(y^i) = P(G_{\phi_i} > a) = 1 - \exp(-\exp(\phi_i - a))$ is the probability that the perturbed log-probability of $y^i$ exceeds $a$. Then we can use the following estimator:

$$\mathbb{E}_{y \sim p_{\boldsymbol{\theta}}(y)}[f(y)] \approx \sum_{i \in S} \frac{p_{\boldsymbol{\theta}}(y^i)}{q_{\boldsymbol{\theta}, \kappa}(y^i)} f(y^i). \tag{5}$$

This estimator is unbiased, and we include a copy of the proof by Kool et al. (2019b) (adapted from the proofs by Duffield et al. (2007) and Vieira (2017)) in Appendix A, as this introduces notation and is the basis for the proof in Appendix C.

Intuition behind this estimator comes from the related *threshold sampling* scenario, where instead of fixing the sample size $k$, we fix the threshold $a$ and define a *variably* sized sample $S = \{i \in \{1, ..., n\} : G_{\phi_i} > a\}$. With threshold sampling, each element $y^i$ in the domain is sampled *independently* with probability $P(G_{\phi_i} > a) = q_{\boldsymbol{\theta}, a}(y^i)$, and $\frac{p_{\boldsymbol{\theta}}(y^i)}{q_{\boldsymbol{\theta}, a}(y^i)}$ is a standard importance weight. As it turns out, instead of having a fixed threshold $a$, we can fix the sample size $k$ and use $\kappa$ as *empirical threshold* (as $i \in S$ if $G_{\phi_i} > \kappa$), and still obtain an unbiased estimator (Duffield et al., 2007; Vieira, 2017).

As was shown by Kool et al. (2019b), in practice it is preferred to normalize the importance weights to reduce variance. This means that we compute the normalization $W(S) = \sum_{i \in S} \frac{p_{\boldsymbol{\theta}}(y^i)}{q_{\boldsymbol{\theta}, \kappa}(y^i)}$ and obtain the following (biased) estimator:

$$\mathbb{E}_{y \sim p_{\boldsymbol{\theta}}(y)}[f(y)] \approx \frac{1}{W(S)} \cdot \sum_{i \in S} \frac{p_{\boldsymbol{\theta}}(y^i)}{q_{\boldsymbol{\theta}, \kappa}(y^i)} f(y^i) \tag{6}$$

## 3 REINFORCE WITH MULTIPLE SAMPLES

Typically REINFORCE is applied with a single sample $y$ per datapoint $x$ (e.g. one translation per source sentence, or, in our experiment, a single tour per TSP instance). In some cases, it may be preferred to take multiple samples $y$ per datapoint $x$ as this requires less data. Taking multiple samples also gives us counterfactual information which can be used to construct a strong (local) baseline. Additionally, we obtain computational benefits, as for encoder-decoder models we can obtain multiple samples using only a single pass through the encoder.

### 3.1 REINFORCE WITH REPLACEMENT

With replacement, we can use the estimator in equation 2, where we can construct a baseline $B_i$ for the $i$-th term based on the other samples $j \neq i$: $B_i = \frac{1}{k-1} \sum_{j \neq i} f(y_j)$. We obtain the following REINFORCE estimator based on drawing $k$ samples $y_1, ..., y_k$ with replacement:

$$\nabla_{\boldsymbol{\theta}} \mathbb{E}_{y \sim p_{\boldsymbol{\theta}}(y)}[f(y)] \approx \frac{1}{k} \sum_{i=1}^{k} \nabla_{\boldsymbol{\theta}} \log p_{\boldsymbol{\theta}}(y_i) \left( f(y_i) - \frac{1}{k-1} \sum_{j \neq i} f(y_j) \right) \tag{7}$$

$$= \frac{1}{k-1} \sum_{i=1}^{k} \nabla_{\boldsymbol{\theta}} \log p_{\boldsymbol{\theta}}(y_i) \left( f(y_i) - \frac{1}{k} \sum_{j=1}^{k} f(y_j) \right). \tag{8}$$

The form in equation 8 is convenient for implementation as it allows to compute a fixed 'baseline' $B = \frac{1}{k}\sum_{j=1}^{k} f(y_j)$ once and correct for the bias (as $B$ depends on $y_i$) by normalizing using $\frac{1}{k-1}$ instead of $\frac{1}{k}$. For details and a proof of unbiasedness we refer to Appendix C.

## 3.2 REINFORCE WITHOUT REPLACEMENT

The basic REINFORCE without replacement estimator follows from combining equation 1 with equation 5 for an unbiased estimator:

$$\nabla_{\boldsymbol{\theta}}\mathbb{E}_{y \sim p_{\boldsymbol{\theta}}(y)}\left[f(y)\right] \approx \sum_{i \in S} \frac{p_{\boldsymbol{\theta}}(y^i)}{q_{\boldsymbol{\theta},\kappa}(y^i)}\nabla_{\boldsymbol{\theta}}\log p_{\boldsymbol{\theta}}(y^i)f(y^i) = \sum_{i \in S} \frac{\nabla_{\boldsymbol{\theta}}p_{\boldsymbol{\theta}}(y^i)}{q_{\boldsymbol{\theta},\kappa}(y^i)}f(y^i) \tag{9}$$

Similar to equation 6, we can compute a lower variance but biased variant by normalizing the importance weights using the normalization $W(S) = \sum_{i \in S} \frac{p_{\boldsymbol{\theta}}(y^i)}{q_{\boldsymbol{\theta},\kappa}(y^i)}$.

When sampling without replacement, the individual samples are *dependent*, and therefore we cannot simply define a baseline based on the other samples as we did in Section 3.1. However, similar to the 'baseline' $\frac{1}{k}\sum_{j=1}^{k} f(y_j)$ in equation 8, we can define an estimate of $\mathbb{E}_{y \sim p_{\boldsymbol{\theta}}(y)}\left[f(y)\right]$ based on the complete sample $S$ (without replacement), using equation 5: $B(S) = \sum_{j \in S} \frac{p_{\boldsymbol{\theta}}(y^j)}{q_{\boldsymbol{\theta},\kappa}(y^j)}f(y^j)$. Using this baseline introduces a bias that we cannot simply correct for by a constant term (as we did in equation 8), as the importance weights depend on $y^i$. Instead, we weight the *individual* terms by $1 - p_{\boldsymbol{\theta}}(y^i) + \frac{p_{\boldsymbol{\theta}}(y^i)}{q_{\boldsymbol{\theta},\kappa}(y^i)}$:

$$\nabla_{\boldsymbol{\theta}}\mathbb{E}_{y \sim p_{\boldsymbol{\theta}}(y)}\left[f(y)\right] \approx \sum_{i \in S} \frac{\nabla_{\boldsymbol{\theta}}p_{\boldsymbol{\theta}}(y^i)}{q_{\boldsymbol{\theta},\kappa}(y^i)}\left(f(y^i)\left(1 - p_{\boldsymbol{\theta}}(y^i) + \frac{p_{\boldsymbol{\theta}}(y^i)}{q_{\boldsymbol{\theta},\kappa}(y^i)}\right) - B(S)\right) \tag{10}$$

This estimator is unbiased and we give the full proof in Appendix C.

For the normalized version, we use the normalization $W(S) = \sum_{i \in S} \frac{p_{\boldsymbol{\theta}}(y^i)}{q_{\boldsymbol{\theta},\kappa}(y^i)}$ for the baseline, and $W_i(S) = W(S) - \frac{p_{\boldsymbol{\theta}}(y^i)}{q_{\boldsymbol{\theta},\kappa}(y^i)} + p_{\boldsymbol{\theta}}(y^i)$ to normalize the outer terms:

$$\nabla_{\boldsymbol{\theta}}\mathbb{E}_{y \sim p_{\boldsymbol{\theta}}(y)}\left[f(y)\right] \approx \sum_{i \in S} \frac{1}{W_i(S)} \cdot \frac{\nabla_{\boldsymbol{\theta}}p_{\boldsymbol{\theta}}(y^i)}{q_{\boldsymbol{\theta},\kappa}(y^i)}\left(f(y^i) - \frac{B(S)}{W(S)}\right) \tag{11}$$

It seems odd to normalize the terms in the outer sum by $\frac{1}{W_i(S)}$ instead of $\frac{1}{W(S)}$, but this estimator can be considered the (normalized) without-replacement equivalent of equation 8 where we normalize by $\frac{1}{k-1}$ instead of $\frac{1}{k}$. It can be derived by rewriting equation 10 to a form similar to equation 7 (as this reveals the actual 'baseline'), then applying the normalization of the importance-weights for the outer sum and the baseline, and then rewriting it once again to the form similar to equation 8 to obtain equation 11. This derivation is given in full in Appendix C.1. The estimator in equation 11 is convenient to implement as the (normalized) 'baseline' $\frac{B(S)}{W(S)}$ only has to be computed once.

## 4 EXPERIMENT

We consider the task of predicting the solution for instances of the Travelling Salesman Problem (TSP) (Vinyals et al., 2015a; Bello et al., 2016; Kool et al., 2019a). The problem is to find the order in which to visit locations (specified by their $x, y$ coordinates) to minimize total travelling distance. A policy is trained using REINFORCE to minimize the expected length of a tour (sequence of locations) predicted by the model.

The Attention Model by Kool et al. (2019a) is a sequence model that considers each instance as a fully connected graph of nodes which are processed by an encoder. The decoder then produces the tour as a sequence of nodes to visit, one node at a time, where it autoregressively uses as input the node visited in the previous step.

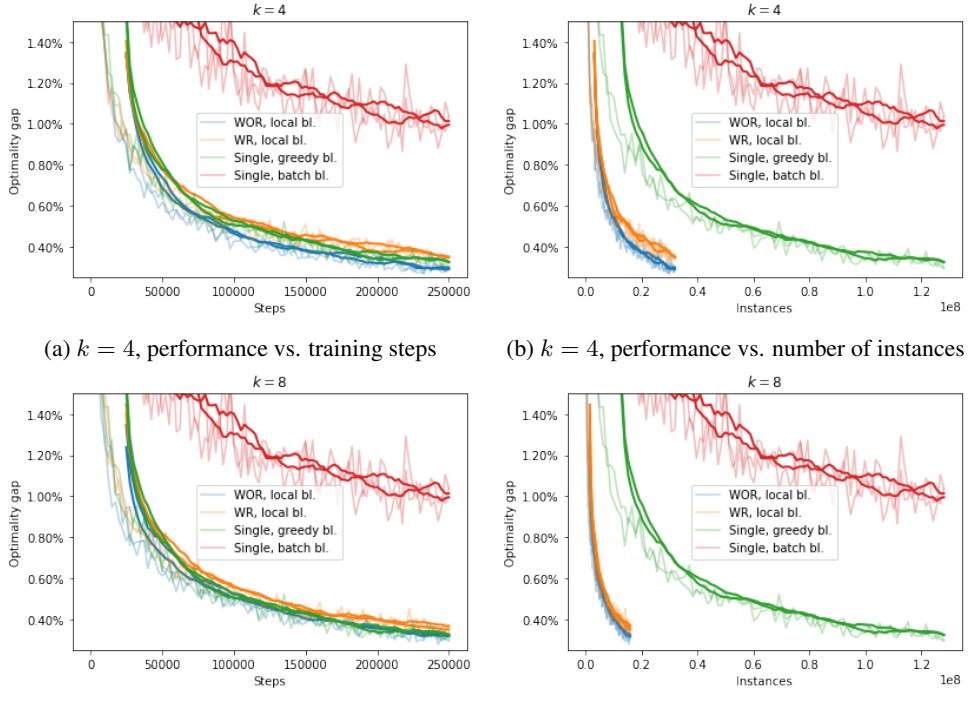

(a) $k = 4$, performance vs. training steps

(b) $k = 4$, performance vs. number of instances

(c) $k = 8$, performance vs. training steps

(d) $k = 8$, performance vs. number of instances

Figure 1: Performance measured as validation set optimality gap during training. Raw results are light, smoothed results are darker (2 random seeds per setting). REINFORCE is used with replacement (WR) and without replacement (WOR) using $k = 4$ (top row) or $k = 8$ (bottom row) samples per instance, and a local baseline based on the $k$ samples for each instance. We compare against REINFORCE using one sample per instance, either with a baseline that is the average of the batch, or the strong greedy rollout baseline by Kool et al. (2019a) that requires an additional rollout of the model.

We use the source code by Kool et al. (2019a)[1] to reproduce their TSP experiment with 20 nodes (as larger instances diminish the benefit of sampling without replacement). We implement REINFORCE estimators based on multiple samples, either sampled with replacement (WR) or without replacement (WOR) using Stochastic Beam Search (Kool et al., 2019b). We compare the following four estimators:

- **Single sample with a batch baseline**. Here we compute the standard REINFORCE estimator (equation 2) with a single sample ($k = 1$). We use a batch of 512 instances (datapoints) and as baseline we take the average of the tour lengths in the *batch*, hence each instances uses the same baseline. This is implemented as using the exponential moving average baseline by (Kool et al., 2019a) with $\beta = 0$.

- **Single sample with a greedy rollout baseline**, and batch size 512. As baseline, we use a *greedy rollout*: for each instance $x$ we take the length of the tour that is obtained by greedily selecting the next location according to an earlier (frozen) version of the model. This baseline, similar to self-critical training (Rennie et al., 2017), corresponds to the best result found by Kool et al. (2019a), superior to using an exponential moving average or learned value function. However, the greedy rollout requires an additional forward pass through the model.

- **Multiple samples with replacement (WR) with a local baseline**. Here we compute the estimator in equation 8 based on $k = 4, 8$ samples. We use a batch size of $\frac{512}{k}$, so the total number of samples is the same. The baseline is local as it is different for each datapoint, but it does not require additional model evaluations like the greedy rollout.

---

[1] https://github.com/wouterkool/attention-learn-to-route

- **Multiple samples without replacement (WOR) with a local baseline**. Here we use the (biased) *normalized* without replacement estimator in equation 11 with $k = 4, 8$ samples and batch size $\frac{512}{k}$. Samples are drawn without replacement using Stochastic Beam Search (Kool et al., 2019b). For fair comparison, we do not take a $(k+1)$-th sample to compute $\kappa$, but sacrifice the $k$-th sample and compute the summation in equation 11 with the remaining $k - 1$ (3 or 7) samples.

Note that a single sample with a local baseline is not possible, which is why we use the batch baseline. The model architecture and training hyperparameters (except batch size) are as in the paper by Kool et al. (2019a). We present the results in terms of the validation set (not used for additional tuning) optimality gap during training in Figure 1, using $k = 4$ (top row) and $k = 8$ (bottom row). We found diminishing returns for larger $k$. The left column presents the results in terms of the number of gradient update steps (minibatches). We see that sampling without replacement performs on par ($k = 8$) or slightly better than using the strong but computationally expensive greedy rollout baseline or using multiple samples with replacement. The standard batch baseline performs significantly worse. The estimators based on multiple samples do not lose (much) final performance, while using significantly less instances. In the right column, where results are presented in terms of the number of instances, this effectiveness is confirmed, and we observe that sampling without replacement is preferred to sampling with replacement. The difference is small, but there is also not much room for improvement as results are close to optimal. The benefit of learning with less data may be small if data is easily generated (as in our setting), but there is also a significant computational benefit as we need significantly fewer encoder evaluations.

## 5 Discussion

In this paper, we have derived REINFORCE estimators based on drawing multiple samples, with and without replacement, and evaluated the effectiveness of the proposed estimators in a structured prediction setting: the prediction of tours for the TSP. The derived estimators yield results comparable to recent results using REINFORCE with a strong greedy rollout baseline, at greater data-efficiency and computational efficiency.

These estimators are especially well suited for structured prediction settings, where the domain is too large to compute exact gradients, but we are able to take multiple samples for the same datapoint, and the objective is a deterministic function of the sampled prediction. We hope the proposed estimators have potential to be used to improve training efficiency in more structured prediction settings, for example in the context of Neural Machine Translation or Image Captioning, where depending on the entropy of the model, sampling without replacement may yield a beneficial improvement.

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

# A    PROOF OF UNBIASEDNESS OF PRIORITY SAMPLING ESTIMATOR

We include here in full the proof by Kool et al. (2019b), as this introduces necessary notation and helps understanding of the proof in Appendix C.

## A.1    PROOF OF UNBIASEDNESS OF PRIORITY SAMPLING ESTIMATOR BY KOOL ET AL. (2019B)

The following proof is adapted from the proofs by Duffield et al. (2007) and Vieira (2017). For generality of the proof, we write $f(i) = f(y^i)$, $p_i = p_{\boldsymbol{\theta}}(y^i)$ and $q_i(\kappa) = q_{\boldsymbol{\theta},\kappa}(y^i)$, and we consider general keys $h_i$ (not necessarily Gumbel perturbations).

We assume we have a probability distribution over a finite domain $1, ..., n$ with normalized probabilities $p_i$, e.g. $\sum_{i=1}^{n} p_i = 1$. For a given function $f(i)$ we want to estimate the expectation

$$\mathbb{E}[f(i)] = \sum_{i=1}^{n} p_i f(i).$$

Each element $i$ has an associated random key $h_i$ and we define $q_i(a) = P(h_i > a)$. This way, if we *know* the threshold $a$ it holds that $q_i(a) = P(i \in S)$ is the probability that element $i$ is in the sample $S$. As was noted by Vieira (2017), the actual distribution of the key does not influence the unbiasedness of the estimator but does determine the effective sampling scheme. Using the Gumbel perturbed log-probabilities as keys (e.g. $h_i = G_{\phi_i}$) is equivalent to the PPSWOR scheme described by Vieira (2017).

We define shorthand notation $h_{1:n} = \{h_1, ..., h_n\}$, $h_{-i} = \{h_1, ..., h_{i-1}, h_{i+1}, ..., h_n\} = h_{1:n} \setminus \{h_i\}$. For a given sample size $k$, let $\kappa$ be the $(k+1)$-th largest element of $h_{1:n}$, so $\kappa$ is the *empirical threshold*. Let $\kappa'_i$ be the $k$-th largest element of $h_{-i}$ (the $k$-th largest of all *other* elements).

Similar to Duffield et al. (2007) we will show that every element $i$ in our sample contributes an unbiased estimate of $\mathbb{E}[f(i)]$, so that the total estimator is unbiased. Formally, we will prove that

$$\mathbb{E}_{h_{1:n}} \left[ \frac{\mathbb{1}_{\{i \in S\}}}{q_i(\kappa)} \right] = 1 \tag{12}$$

from which the result follows:

$$
\begin{aligned}
&\mathbb{E}_{h_{1:n}} \left[ \sum_{i \in S} \frac{p_i}{q_i(\kappa)} f(i) \right] \\
&= \mathbb{E}_{h_{1:n}} \left[ \sum_{i=1}^{n} \frac{p_i}{q_i(\kappa)} f(i) \mathbb{1}_{\{i \in S\}} \right] \\
&= \sum_{i=1}^{n} p_i f(i) \cdot \mathbb{E}_{h_{1:n}} \left[ \frac{\mathbb{1}_{\{i \in S\}}}{q_i(\kappa)} \right] \\
&= \sum_{i=1}^{n} p_i f(i) \cdot 1 = \sum_{i=1}^{n} p_i f(i) = \mathbb{E}[f(i)]
\end{aligned}
$$

To prove equation 12, we make use of the observation (slightly rephrased) by Duffield et al. (2007) that conditioning on $h_{-i}$, we know $\kappa'_i$ and the event $i \in S$ implies that $\kappa = \kappa'_i$ since $i$ will only be in the sample if $h_i > \kappa'_i$ which means that $\kappa'_i$ is the $k+1$-th largest value of $h_{-i} \cup \{h_i\} = h_{1:n}$. The reverse is also true (if $\kappa = \kappa'_i$ then $h_i$ must be larger than $\kappa'_i$ since otherwise the $k+1$-th largest

value of $h_{1:n}$ will be smaller than $\kappa_i'$).

$$
\mathbb{E}_{h_{1:n}} \left[ \frac{\mathbb{1}_{\{i \in S\}}}{q_i(\kappa)} \right]
$$

$$
= \mathbb{E}_{h_{-i}} \left[ \mathbb{E}_{h_i} \left[ \frac{\mathbb{1}_{\{i \in S\}}}{q_i(\kappa)} \middle| h_i \right] \right]
$$

$$
= \mathbb{E}_{h_{-i}} \left[ \mathbb{E}_{h_i} \left[ \frac{\mathbb{1}_{\{i \in S\}}}{q_i(\kappa)} \middle| h_{-i}, i \in S \right] P(i \in S | h_{-i}) + \mathbb{E}_{h_i} \left[ \frac{\mathbb{1}_{\{i \in S\}}}{q_i(\kappa)} \middle| h_{-i}, i \notin S \right] P(i \notin S | h_{-i}) \right]
$$

$$
= \mathbb{E}_{h_{-i}} \left[ \mathbb{E}_{h_i} \left[ \frac{1}{q_i(\kappa)} \middle| h_{-i}, i \in S \right] P(i \in S | h_{-i}) + 0 \right]
$$

$$
= \mathbb{E}_{h_{-i}} \left[ \mathbb{E}_{h_i} \left[ \frac{1}{q_i(\kappa)} \middle| h_{-i}, i \in S \right] q_i(\kappa_i') \right]
$$

$$
= \mathbb{E}_{h_{-i}} \left[ \mathbb{E}_{h_i} \left[ \frac{1}{q_i(\kappa)} \middle| \kappa = \kappa_i' \right] q_i(\kappa_i') \right]
$$

$$
= \mathbb{E}_{h_{-i}} \left[ \mathbb{E}_{h_i} \left[ \frac{1}{q_i(\kappa_i')} \right] q_i(\kappa_i') \right]
$$

$$
= \mathbb{E}_{h_{-i}} \left[ \frac{1}{q_i(\kappa_i')} q_i(\kappa_i') \right] = \mathbb{E}_{h_{-i}} [1] = 1
$$

## B  REINFORCE WITH BASELINE AND REPLACEMENT

We will now prove that the REINFORCE estimator based on multiple samples with the sample average as baseline (equation 8) is unbiased. Let $y_{1:k} = \{y_1, ..., y_k\}$ be the set of *independent* samples (with replacement) from $p_{\boldsymbol{\theta}}(y)$. First we show that using the batch mean as baseline is equivalent to using the mean of the other elements in the batch, up to a constant $\frac{k-1}{k}$.

$$
f(y_i) - \frac{1}{k} \sum_{j=1}^{k} f(y_j)
$$

$$
= f(y_i)(1 - \frac{1}{k}) - \frac{1}{k} \sum_{j \neq i} f(y_j)
$$

$$
= f(y_i) \frac{k-1}{k} - \frac{1}{k-1} \cdot \frac{k-1}{k} \sum_{j \neq i} f(y_j)
$$

$$
= \frac{k-1}{k} \left( f(y_i) - \frac{1}{k-1} \cdot \sum_{j \neq i} f(y_j) \right) \tag{13}
$$

Note that $\frac{k-1}{k}$ goes to 1 as the batch size $k$ increases and we do not need to include it (and we can simply compute the biased mean) as it can be absorbed into the learning rate. Since $y_j$ is independent

of $y_i$, unbiasedness follows:

$$\mathbb{E}_{y_{1:k}}\left[\frac{1}{k-1}\sum_{i=1}^{k}\nabla_{\boldsymbol{\theta}}\log p_{\boldsymbol{\theta}}(y_i)\left(f(y_i)-\frac{1}{k}\sum_{j=1}^{k}f(y_j)\right)\right]$$

$$=\mathbb{E}_{y_{1:k}}\left[\frac{1}{k}\sum_{i=1}^{k}\nabla_{\boldsymbol{\theta}}\log p_{\boldsymbol{\theta}}(y_i)\left(f(y_i)-\frac{1}{k-1}\sum_{j\neq i}f(y_j)\right)\right]$$

$$=\frac{1}{k}\sum_{i=1}^{k}\mathbb{E}_{y_{1:k}}\left[\nabla_{\boldsymbol{\theta}}\log p_{\boldsymbol{\theta}}(y_i)f(y_i)\right]-\frac{1}{k-1}\sum_{j\neq i}\mathbb{E}_{y_{1:k}}\left[\nabla_{\boldsymbol{\theta}}\log p_{\boldsymbol{\theta}}(y_i)f(y_j)\right]$$

$$=\frac{1}{k}\sum_{i=1}^{k}\mathbb{E}_{y_i}\left[\nabla_{\boldsymbol{\theta}}\log p_{\boldsymbol{\theta}}(y_i)f(y_i)\right]-\frac{1}{k-1}\sum_{j\neq i}\mathbb{E}_{y_i}\left[\nabla_{\boldsymbol{\theta}}\log p_{\boldsymbol{\theta}}(y_i)\right]\cdot\mathbb{E}_{y_j}\left[f(y_j)\right]$$

$$=\frac{1}{k}\sum_{i=1}^{k}\mathbb{E}_{y}\left[\nabla_{\boldsymbol{\theta}}\log p_{\boldsymbol{\theta}}(y)f(y)\right]-\frac{1}{k-1}\sum_{j\neq i}0\cdot\mathbb{E}_{y_j}\left[f(y_j)\right]$$

$$=\frac{1}{k}\cdot k\cdot\mathbb{E}_{y}\left[\nabla_{\boldsymbol{\theta}}\log p_{\boldsymbol{\theta}}(y)f(y)\right]-0$$

$$=\mathbb{E}_{y}\left[\nabla_{\boldsymbol{\theta}}\log p_{\boldsymbol{\theta}}(y)f(y)\right]$$

$$=\nabla_{\boldsymbol{\theta}}\mathbb{E}_{y}\left[f(y)\right]$$

## C  REINFORCE WITH BASELINE WITHOUT REPLACEMENT

The proof that the REINFORCE estimator based on multiple samples *without* replacement with baseline (equation 10) is unbiased follows from adapting and combining the proofs in Appendix A and B. Additionally to $q_i(a) = P(h_i > a)$ we define $q_{ij}(a) = P(h_i > a \cap h_j > a) = P(h_i > a)P(h_j > a) = q_i(a)q_j(a)$ for $i \neq j$ and $q_{ii}(a) = P(h_i > a) = q_i(a)$. For convenience we define shorthand for the *conditional* $q_{j|i}(a) = \frac{q_{ij}(a)}{q_i(a)}$, so $q_{j|i}(a) = q_j(a)$ for $j \neq i$ and $q_{i|i}(a) = 1$.

Furthermore, we define $h_{-ij} = h_{1:n} \setminus \{h_i, h_j\}$ and define $\kappa'_{ij}$ $(i \neq j)$ as the $(k-1)$-th (not $k$-th!) largest element of $h_{-ij}$, and $\kappa'_{ii} = \kappa'_i$, e.g. the $k$-th largest element of $h_{-i}$.

We denote with with $\{i, j \in S\} = \{i \in S \cap j \in S\}$ the event that both $i$ and $j$ are in the sample, also for $i = j$ which simply means $\{i \in S\}$. First we generalize $P(i \in S | h_{-i}) = q_i(\kappa'_i)$ to the pairwise conditional inclusion probability $P(i, j \in S | h_{-ij})$.

**Lemma 1.** $P(i, j \in S | h_{-ij}) = q_{ij}(\kappa'_{ij})$

*Proof.* For $i = j$:

$$P(i, j \in S | h_{-ij}) = P(i \in S | h_{-i}) = q_i(\kappa'_i) =$$
$$q_{ii}(\kappa'_{ii}) = q_{ij}(\kappa'_{ij})$$

For $i \neq j$: Assuming w.l.o.g. $h_i < h_j$ there are the following scenario's:

- $\kappa'_{ij} < h_i < h_j$. In this case, after adding $h_i$ and $h_j$ to $h_{-ij}$, $\kappa'_{ij}$ will be the $(k+1)$-th largest element so $\kappa = \kappa'_{ij}$ and $i \in S$ and $j \in S$ since $h_j > h_i > \kappa = \kappa'_{ij}$.

- $h_i < \kappa'_{ij} < h_j$ or $h_i < h_j < \kappa'_{ij}$. In both cases, there are at least $(k-1)+1 = k$ elements higher than $h_i$ so $i \notin S$.

Therefore it follows that $\{i, j \in S | u_{-ij}\} = \{h_i > \kappa'_{ij} \cap h_j > \kappa'_{ij} | u_{-ij}\}$ and additionally this event implies $\kappa = \kappa'_{ij}$. Now the result follows:

$$
\begin{aligned}
&P(i, j \in S | u_{-ij}) \\
=&P(h_i > \kappa \cap h_j > \kappa | u_{-ij}) \\
=&P(h_i > \kappa'_{ij} \cap h_j > \kappa'_{ij} | u_{-ij}) \\
=&P(h_i > \kappa'_{ij} | u_{-ij}) P(h_j > \kappa'_{ij} | u_{-ij}) \\
=&q_i(\kappa'_{ij}) q_j(\kappa'_{ij}) = q_{ij}(\kappa'_{ij})
\end{aligned}
$$

$\square$

Using this Lemma we can prove the following Lemma:

**Lemma 2.**

$$
\mathbb{E}_{h_{1:n}} \left[ \frac{\mathbb{1}_{\{i,j \in S\}}}{q_{ij}(\kappa)} \right] = 1 \tag{14}
$$

Note that the expectation is w.r.t. the keys $h_{1:n}$ which define the random variables $\kappa$ and $S = \{i : h_i > \kappa\}$.

*Proof.*

$$
\begin{aligned}
&\mathbb{E}_{h_{1:n}} \left[ \frac{\mathbb{1}_{\{i,j \in S\}}}{q_{ij}(\kappa)} \right] \\
=&\mathbb{E}_{h_{-ij}} \left[ \mathbb{E}_{h_i, h_j} \left[ \frac{\mathbb{1}_{\{i,j \in S\}}}{q_{ij}(\kappa)} \bigg| u_{-ij} \right] \right] \\
=&\mathbb{E}_{h_{-ij}} \left[ \mathbb{E}_{h_i, h_j} \left[ \frac{1}{q_{ij}(\kappa)} \bigg| u_{-ij}, i, j \in S \right] P(i, j \in S | u_{-ij}) + 0 \cdot (1 - P(i, j \in S | u_{-ij})) \right] \\
=&\mathbb{E}_{h_{-ij}} \left[ \mathbb{E}_{h_i, h_j} \left[ \frac{1}{q_{ij}(\kappa)} \bigg| \kappa = \kappa'_{ij} \right] q_{ij}(\kappa'_{ij}) \right] \\
=&\mathbb{E}_{h_{-ij}} \left[ \frac{1}{q_{ij}(\kappa'_{ij})} q_{ij}(\kappa'_{ij}) \right] = \mathbb{E}_{h_{-ij}} [1] = 1
\end{aligned}
$$

$\square$

**Theorem 1.** *Let* $B(S) = \sum_{j \in S} \frac{p_{\boldsymbol{\theta}}(y^j)}{q_j(\kappa)} f(y^j)$. *Then the following is an unbiased estimator:*

$$
\mathbb{E}_{h_{1:n}} \left[ \sum_{i \in S} \frac{\nabla_{\boldsymbol{\theta}} p_{\boldsymbol{\theta}}(y^i)}{q_i(\kappa)} \left( f(y^i) \left( 1 - p_{\boldsymbol{\theta}}(y^i) + \frac{p_{\boldsymbol{\theta}}(y^i)}{q_i(\kappa)} \right) - B(S) \right) \right] = \nabla_{\boldsymbol{\theta}} \mathbb{E}_{y \sim p_{\boldsymbol{\theta}}(y)} [f(y)] \tag{15}
$$

*Proof.* First note that, when $i \in S$, we can rewrite:

$$
\begin{aligned}
&f(y^i) \left( 1 - p_{\boldsymbol{\theta}}(y^i) + \frac{p_{\boldsymbol{\theta}}(y^i)}{q_i(\kappa)} \right) - B(S) \\
=&f(y^i) \left( 1 - p_{\boldsymbol{\theta}}(y^i) + \frac{p_{\boldsymbol{\theta}}(y^i)}{q_i(\kappa)} \right) - \sum_{j \in S} \frac{p_{\boldsymbol{\theta}}(y^j)}{q_j(\kappa)} f(y^j) \\
=&f(y^i) (1 - p_{\boldsymbol{\theta}}(y^i)) - \sum_{j \in S \setminus \{i\}} \frac{p_{\boldsymbol{\theta}}(y^j)}{q_j(\kappa)} f(y^j) \\
=&f(y^i) \left( 1 - \frac{p_{\boldsymbol{\theta}}(y^i)}{q_{i|i}(\kappa)} \right) - \sum_{j \in S \setminus \{i\}} \frac{p_{\boldsymbol{\theta}}(y^j)}{q_{j|i}(\kappa)} f(y^j) \\
=&f(y^i) - \sum_{j \in S} \frac{p_{\boldsymbol{\theta}}(y^j)}{q_{j|i}(\kappa)} f(y^j) \tag{16}
\end{aligned}
$$

Then the proof follows:

$$\mathbb{E}_{h_{1:n}}\left[\sum_{i\in S}\frac{\nabla_{\boldsymbol{\theta}}p_{\boldsymbol{\theta}}(y^i)}{q_i(\kappa)}\left(f(y^i)\left(1-p_{\boldsymbol{\theta}}(y^i)+\frac{p_{\boldsymbol{\theta}}(y^i)}{q_i(\kappa)}\right)-B(S)\right)\right]$$

$$=\mathbb{E}_{h_{1:n}}\left[\sum_{i\in S}\frac{\nabla_{\boldsymbol{\theta}}p_{\boldsymbol{\theta}}(y^i)}{q_i(\kappa)}\left(f(y^i)-\sum_{j\in S}\frac{p_{\boldsymbol{\theta}}(y^j)}{q_{j|i}(\kappa)}f(y^j)\right)\right]$$

$$=\mathbb{E}_{h_{1:n}}\left[\sum_{i=1}^{n}\mathbb{1}_{\{i\in S\}}\frac{\nabla_{\boldsymbol{\theta}}p_{\boldsymbol{\theta}}(y^i)}{q_i(\kappa)}\left(f(y^i)-\sum_{j=1}^{n}\mathbb{1}_{\{j\in S\}}\frac{p_{\boldsymbol{\theta}}(y^j)}{q_{j|i}(\kappa)}f(y^j)\right)\right]$$

$$=\sum_{i=1}^{n}\nabla_{\boldsymbol{\theta}}p_{\boldsymbol{\theta}}(y^i)\cdot\mathbb{E}_{h_{1:n}}\left[\frac{\mathbb{1}_{\{i\in S\}}}{q_i(\kappa)}\left(f(y^i)-\sum_{j=1}^{n}\frac{\mathbb{1}_{\{j\in S\}}}{q_{j|i}(\kappa)}p_{\boldsymbol{\theta}}(y^j)f(y^j)\right)\right]$$

$$=\sum_{i=1}^{n}\nabla_{\boldsymbol{\theta}}p_{\boldsymbol{\theta}}(y^i)\cdot\mathbb{E}_{h_{1:n}}\left[\frac{\mathbb{1}_{\{i\in S\}}}{q_i(\kappa)}f(y^i)-\sum_{j=1}^{n}\frac{\mathbb{1}_{\{i,j\in S\}}}{q_{ij}(\kappa)}p_{\boldsymbol{\theta}}(y^j)f(y^j)\right]$$

$$=\sum_{i=1}^{n}\nabla_{\boldsymbol{\theta}}p_{\boldsymbol{\theta}}(y^i)\left(f(y^i)\cdot\mathbb{E}_{h_{1:n}}\left[\frac{\mathbb{1}_{\{i\in S\}}}{q_i(\kappa)}\right]-\sum_{j=1}^{n}p_{\boldsymbol{\theta}}(y^j)f(y^j)\cdot\mathbb{E}_{h_{1:n}}\left[\frac{\mathbb{1}_{\{i,j\in S\}}}{q_{ij}(\kappa)}\right]\right)$$

$$=\sum_{i=1}^{n}\nabla_{\boldsymbol{\theta}}p_{\boldsymbol{\theta}}(y^i)\left(f(y^i)\cdot 1-\sum_{j=1}^{n}p_{\boldsymbol{\theta}}(y^j)f(y^j)\cdot 1\right)$$

$$=\sum_{i=1}^{n}\nabla_{\boldsymbol{\theta}}p_{\boldsymbol{\theta}}(y^i)\left(f(y^i)-\mathbb{E}_{y\sim p_{\boldsymbol{\theta}}(y)}[f(y)]\right)$$

$$=\sum_{i=1}^{n}\nabla_{\boldsymbol{\theta}}p_{\boldsymbol{\theta}}(y^i)f(y^i)-\sum_{i=1}^{n}\nabla_{\boldsymbol{\theta}}p_{\boldsymbol{\theta}}(y^i)\mathbb{E}_{y\sim p_{\boldsymbol{\theta}}(y)}[f(y)]$$

$$=\sum_{i=1}^{n}\nabla_{\boldsymbol{\theta}}p_{\boldsymbol{\theta}}(y^i)f(y^i)-\mathbb{E}_{y\sim p_{\boldsymbol{\theta}}(y)}[f(y)]\cdot\sum_{i=1}^{n}\nabla_{\boldsymbol{\theta}}p_{\boldsymbol{\theta}}(y^i)$$

$$=\nabla_{\boldsymbol{\theta}}\sum_{i=1}^{n}p_{\boldsymbol{\theta}}(y^i)f(y^i)-\mathbb{E}_{y\sim p_{\boldsymbol{\theta}}(y)}[f(y)]\cdot\nabla_{\boldsymbol{\theta}}\sum_{i=1}^{n}p_{\boldsymbol{\theta}}(y^i)$$

$$=\nabla_{\boldsymbol{\theta}}\mathbb{E}_{y\sim p_{\boldsymbol{\theta}}(y)}[f(y)]-\mathbb{E}_{y\sim p_{\boldsymbol{\theta}}(y)}[f(y)]\cdot\nabla_{\boldsymbol{\theta}}1$$

$$=\nabla_{\boldsymbol{\theta}}\mathbb{E}_{y\sim p_{\boldsymbol{\theta}}(y)}[f(y)]-\mathbb{E}_{y\sim p_{\boldsymbol{\theta}}(y)}[f(y)]\cdot 0$$

$$=\nabla_{\boldsymbol{\theta}}\mathbb{E}_{y\sim p_{\boldsymbol{\theta}}(y)}[f(y)]$$

$\square$

## C.1 Normalized importance weights

By rewriting $\frac{\nabla_{\boldsymbol{\theta}}p_{\boldsymbol{\theta}}(y^i)}{q_i(\kappa)}=\frac{p_{\boldsymbol{\theta}}(y^i)}{q_i(\kappa)}\cdot\nabla_{\boldsymbol{\theta}}\log p_{\boldsymbol{\theta}}$ we reveal that the estimator equation 15 is actually an importance weighted REINFORCE estimator. Although unbiased, in practice it is preferred to use the *normalized* importance weight estimator. We will therefore normalize both the outer weights as well as the weights used for computation of the baseline. For this, write $W(S)=\sum_{i\in S}\frac{p_{\boldsymbol{\theta}}(y^i)}{q_i(\kappa)}$ and scale equation 15 by $\frac{1}{W(S)}$. Note that although equation 15 is written in terms of $B(S)$, the actual baseline (for sample $i$) is $\sum_{j\in S}\frac{p_{\boldsymbol{\theta}}(y^j)}{q_{j|i}(\kappa)}f(y^j)$ (see equation 16) which should be normalized by

$$W_i(S)=\sum_{j\in S}\frac{p_{\boldsymbol{\theta}}(y^j)}{q_{j|i}(\kappa)}=W(S)-\frac{p_{\boldsymbol{\theta}}(y^i)}{q_i(\kappa)}+p_{\boldsymbol{\theta}}(y^i).$$

Using equation 16, we can rewrite (similar to equation 13)

$$f(y^i) - \frac{1}{W_i(S)} \sum_{j \in S} \frac{p_{\boldsymbol{\theta}}(y^j)}{q_{j|i}(\kappa)} f(y^j)$$

$$= \frac{1}{W_i(S)} \left( f(y^i)W_i(S) - \sum_{j \in S} \frac{p_{\boldsymbol{\theta}}(y^j)}{q_{j|i}(\kappa)} f(y^j) \right)$$

$$= \frac{1}{W_i(S)} \left( f(y^i) \left( W_i(S) - p_{\boldsymbol{\theta}}(y^i) + \frac{p_{\boldsymbol{\theta}}(y^i)}{q_i(\kappa)} \right) - B(S) \right)$$

$$= \frac{1}{W_i(S)} \left( f(y^i)W(S) - B(S) \right)$$

$$= \frac{W(S)}{W_i(S)} \left( f(y^i) - \frac{B(S)}{W(S)} \right)$$

Substituting this into equation 15 and normalizing the outer importance weights by $W(S)$ we see that this term cancels to obtain

$$\mathbb{E}_{h_{1:n}} \left[ \frac{1}{W(S)} \sum_{i \in S} \frac{\nabla_{\boldsymbol{\theta}} p_{\boldsymbol{\theta}}(y^i)}{q_i(\kappa)} \left( f(y^i) - \frac{1}{W_i(S)} \sum_{j \in S} \frac{p_{\boldsymbol{\theta}}(y^j)}{q_{j|i}(\kappa)} f(y^j) \right) \right]$$

$$= \mathbb{E}_{h_{1:n}} \left[ \frac{1}{W(S)} \sum_{i \in S} \frac{\nabla_{\boldsymbol{\theta}} p_{\boldsymbol{\theta}}(y^i)}{q_i(\kappa)} \cdot \frac{W(S)}{W_i(S)} \left( f(y^i) - \frac{B(S)}{W(S)} \right) \right]$$

$$= \mathbb{E}_{h_{1:n}} \left[ \sum_{i \in S} \frac{1}{W_i(S)} \cdot \frac{\nabla_{\boldsymbol{\theta}} p_{\boldsymbol{\theta}}(y^i)}{q_i(\kappa)} \left( f(y^i) - \frac{B(S)}{W(S)} \right) \right]$$

