# OpenReview forum: "Buy 4 REINFORCE Samples, Get a Baseline for Free!"
_ICLR.cc/2019/Workshop/drlStructPred — drlStructPred 2019_

### Official Review · AnonReviewer4 · 2019-04-05
**Multi-sample REINFORCE estimators w/ and w/o replacement**

**Rating:** 4
**Confidence:** 2

**Review:**

The authors derive a REINFORCE estimator based on sampling without replacement (building on the recent work of Kool et al.) and show improvements over existing techniques on structured prediction.

The paper is well-written and the experiments are informative. The paper could be improved by theoretically quantifying the improvement gain from sampling w/ replacement. This would help to understand why the benefits of sampling without replacement diminish with larger instances and larger k.

---

### Official Review · AnonReviewer5 · 2019-04-07
**Novelty and experiments**

**Rating:** 2
**Confidence:** 2

**Review:**

This work aims to improve REINFORCE, which is used to train models in structured prediction settings to directly optimize the test-time objective. It shows that by drawing multiple samples (predictions) per datapoint, it can learn with significantly less data, as it freely obtains a REINFORCE baseline to reduce variance.

Similar ideas, drawing multiple samples (predictions) per datapoint, have already been studied in RL applications (see below). Those related works are not reviewed in this paper. It is better to discuss the difference between this work and existing works and highlight the novelty of this work.
Minimum Risk Training for Neural Machine Translation, ACL 2016
Dual Learning for Machine Translation, NIPS 2016

The experiments can be improved from several aspects.
1. Compare with recent methods, including non-RL methods such as pointer networks.
2. Only very small scale settings are tested (e.g., 20 nodes). It is better to consider larger scale experiments. For example, pointer networks are tested with 50 nodes.

---

### Official Review · AnonReviewer1 · 2019-04-07
**Unbiased REINFORCE estimator using sampling without replacement**

**Rating:** 4
**Confidence:** 2

**Review:**

In this work the authors suggest to draw several samples without replacement in REINFORCE training, allowing them to use a local baseline, i.e. a baseline with regard to a single instance as opposed to computing the baseline over the other examples in the batch. Some complexity comes from the fact that the multi-sample-baseline is much easier derived when instead sampling with replacement as common in related work.

This paper relies heavily on previous work that established the "Gumbel top-k trick" which allows sampling without replacement, but self-sufficient given the references. Their new contribution is a gradient estimator based on sampling without replacement, including a variant with a baseline that is based on the other samples produced for the same example (e.g. different translations for the same source sentence).

Experiments on a TSP solver show that sampling several times per input example performs better than doing a single sample and on-par with a computationally expensive version where roll-out is used to produce a baseline for the single sample case. Furthermore sampling without replacement seems to work slightly better than with replacement. That being said, the latter seems much easier to implement.

The proposed approach might be a good fit for the somewhat nice setting where one trains on few examples and samples from a peaked distribution where sampling the same sequence repeatedly would be an issue.

Overall this paper is very pleasant to read and presents the flow of ideas well.

The work is accompanied by a detailed proof that the derived estimator is unbiased. Skimming the proof revealed no issues but I could have missed something here.

---

### Official Review · AnonReviewer2 · 2019-04-08
**REINFORCE loss w/o replacement**

**Rating:** 4
**Confidence:** 2

**Review:**

This paper proposes a method for learning a policy based on a variance reduced REINFORCE loss based on sampling without replacement. Experiments suggest that learning with this approach performs slightly better when compared with a greedy rollout baseline (Kool et al 2019) but it is more sample efficient requiring fewer instances to learn.
The paper provides results in a toy problem using a Travelling Salesman Problem with a few nodes (20), it would be interesting to evaluate the performance on a more complex task to see whether sampling without replacement would still be beneficial over other sampling schemes.
It would also be interesting to evaluate the performance of the normalized and non-normalized version of the loss, to see how much variance is reduce between the two proposed approaches.

This paper provides interesting preliminary results on an important topic for deep structure prediction with RL and therefore I would like to see it presented in this workshop.

---

### Decision · Program_Chairs · 2019-04-09
**Acceptance Decision**

Accept